# Caspase-11 Non-Canonical Inflammasome: Emerging Activator and Regulator of Infection-Mediated Inflammatory Responses

**DOI:** 10.3390/ijms21082736

**Published:** 2020-04-15

**Authors:** Young-Su Yi

**Affiliations:** Department of Life Sciences, Kyonggi University, Suwon 16227, Korea; ysyi@kgu.ac.kr; Tel.: +82-31-249-9644

**Keywords:** caspase-11, infection, inflammasome, LPS, non-canonical

## Abstract

Inflammation is a body’s protective mechanism to eliminate invading pathogens and cellular damaging signals. The inflammatory response consists of two main consecutive steps—a priming step preparing the inflammatory responses and a triggering step boosting the inflammatory responses. The main feature of the triggering step is the activation of the inflammasome, an intracellular multiprotein complex facilitating the inflammatory responses. The regulatory roles of ‘canonical’ inflammasomes in the inflammatory responses and diseases have been largely investigated, so far. New types of inflammasomes have been recently discovered and named as ‘non-canonical’ inflammasomes since their roles to induce inflammatory responses are similar to those of canonical inflammasomes, however, the stimulating ligands and the underlying mechanisms are different. Therefore, a growing number of studies have actively investigated the novel roles of non-canonical inflammasomes in inflammatory responses and diseases. This review summarizes and discusses the recent studies exploring the regulatory roles of caspase-11 non-canonical inflammasome during the inflammatory responses and provides insight into the development of novel therapeutics for infectious and inflammatory diseases by targeting caspase-11 non-canonical inflammasome.

## 1. Introduction

Inflammation is a series of innate immune responses to protect the body from various invading pathogens and cellular danger stress signals [1,2]. Although inflammation is a body-protective mechanism, chronic inflammation and repeated and prolonged inflammation are considered as risk factors for various human diseases, such as inflammatory diseases, autoimmune diseases, and cancers [3,4,5,6]. The inflammatory response consists of two main steps—priming and triggering.

The priming step is a preparation for increasing the expression of the inflammatory mediators, such as inflammatory molecules and pro-inflammatory cytokines. Priming is initiated by the interaction between extracellular pattern recognition receptors (PRRs), such as Toll-like receptors (TLRs), the pathogen-associated molecular patterns (PAMPs), and the danger-associated molecular patterns (DAMPs) [1,2]. This interaction, in turn, induces the signal transduction cascades of intracellular molecules in the inflammatory signaling pathways, such as nuclear factor-kappa B (NF-κB), activator protein-1 (AP-1), and interferon (IFN) regulatory factors (IRFs), resulting in the production of inflammatory molecules, such as nitric oxide (NO) and prostaglandin E_2_ (PGE_2_) and pro-inflammatory cytokines, tumor necrosis factor (TNF)-α, interleukin (IL)-1Β, IL-6, and IFNs [7,8,9,10].

The triggering step, on the other hand, is initiated by the interaction of intracellular PRRs, such as the nucleotide-binding oligomerization domain-like receptors (NLRs), retinoic acid-inducible gene-I-like receptors, absent in melanoma 2, absent in melanoma 2-like receptors, caspase-11, and caspase-4 with a variety of PAMPs and DAMPs [11,12]. This interaction subsequently forms an inflammatory protein complex (inflammasome) and induces the caspase-1-mediated proteolytic maturation of pro-inflammatory cytokines, IL-1β and IL-18 as well as the gasdermin D (GSDMD)-mediated formation of membrane pores and pyroptosis (an inflammatory form of cell death) [11,12,13,14].

The inflammasome is categorized into two main groups: The first category is canonical inflmmasomes that include absent in melanoma 2 inflammasome, pyrin inflammasome, and NLR family inflammasomes, such as NLRP1, NLRP3, NLRC4, and NLRP6 inflammasomes. Canonical inflammasomes were discovered earlier, and a large number of studies have demonstrated that they play a pivotal role in inflammatory responses and the pathogenesis of various inflammatory diseases [15,16,17]. A new class of inflammasomes with roles similar to those of canonical inflammasomes was unexpectedly discovered, however, the stimulating PAMPs and the underlying mechanism of inflammatory responses mediated by these inflammasomes is different from the canonical inflammasomes. Therefore, these new inflammasomes were named ‘non-canonical’ inflammasomes and include mouse caspase-11 and human caspase-4 and -5 that are considered as homologues of mouse caspase-11 [18,19].

Since the non-canonical inflammasomes were recently discovered, limited numbers of studies investigating their roles in the inflammatory responses and diseases have been reported. However, recent emerging studies have reported new regulatory roles of non-canonical inflammasomes in the inflammatory responses. Therefore, this review aims to discuss the recent progress in understanding the roles of non-canonical inflammasomes, especially focusing on caspase-11 non-canonical inflammasome in the inflammatory responses since caspase-11 non-canonical inflammasome was the first discovered and the most studied. Additionally, this review highlights the promising strategy for developing novel therapeutics for the prevention and treatment of infectious and inflammatory diseases by selectively targeting caspase-11 non-canonical inflammasome.

## 2. Activation of Caspase-11 Non-Canonical Inflammasome

### 2.1. Structure

Caspases are a family of cysteine aspartate-specific endogenous proteases. Some caspases consist of a N-terminal caspase recruit domain (CARD), a middle p20 (large) domain, and a C-terminal p10 (small) domain, while other caspases consist of only p20 and p10 domains and lack an N-terminal CARD [20]. Caspase-8 and -10 have an N-terminal death effector domain instead of CARD [20]. Caspase-11 consists of an N-terminal CARD, a p20 domain, and a C-terminal p10 domain and is 373 amino acids in length (Figure 1A). Caspase-11 was first discovered in mice [18], and many efforts were made to identify the same protein in humans, with no success. Instead, two human homologs of mouse caspase-11, caspase-4 and -5 were identified [19]. Like caspase-11, human caspase-4 and -5 also consist of CARD, p20, and p10 domains (Figure 1A), but they differ from caspase-11 in size. Caspase-4 and -5 are 377 and 434 amino acids in length, respectively (Figure 1A), which might be a clue that the human counterpart of mouse caspase-11 is caspase-4 rather than caspase-5. However, further studies need to be carried out to determine which one is more similar in function to caspase-11.

### 2.2. Activators

Similarly to canonical inflammasomes, caspase-11 non-canonical inflammasome is also activated in response to stimulating molecules. Lipopolysaccharide (LPS) was the first molecule identified to activate caspase-11 non-canonical inflammasome by direct interaction. LPS is a cell wall component and endotoxin derived from Gram-negative bacteria consisting of three structural domains: lipid A, core, and O antigen. LPS activates caspase-11 non-canonical inflammasome by interaction between lipid A and CARD of caspase-11 (Figure 1B

caspase-11, caspase-4/5 also interact with LPS, leading to the activation of caspase-4/5 non-canonical inflammasomes [19]. The recognition of LPS by caspase-4/5/11 then results in the activation of caspase-4/5/11 non-canonical inflammasomes via the oligomerization of LPS-caspase-4/5/11 complexes (Figure 1C).

Various efforts have been made to discover new ligands to activate caspase-11 non-canonical inflammasome. Lipophosphoglycan (LPG) is a glycolipid in the cell walls of single-cell protozoan *Leishmania* parasites that cause Leishmaniasis in humans and other mammals [21]. A recent study reported that LPG of *Leishmania* parasites activated caspase-11 non-canonical inflammasome in macrophages, and that the activation did not happen in the macrophages infected with *lpg1*^−/−^
*Leishmania* parasites [22]. However, this study did not demonstrate the mechanism by which LPG activates caspase-11 non-canonical inflammasome, and therefore, further studies demonstrating the mechanism of LPG-activated caspase-11 non-canonical inflammasome are required.

Another study reported the oxidized form of endogenous phospholipids, 1-palmitoyl-2-arachidonoyl-sn-glycero-3-phosphorylcholine (oxPAPC) as an activator of the caspase-11 non-canonical inflammasome. oxPAPC directly interacted with caspase-11 in dendritic cells, however, unlike LPS, oxPAPC interacted with the catalytic domain of caspase-11 [23], which provided the clue that oxPAPC might activate caspase-11 non-canonical inflammasome differently compared to LPG. In contrast to the previous study [23], another study reported that oxPAPC competed for the interaction of LPS with caspase-11, consequently resulting in the inhibition of caspase-11 non-canonical inflammasome in macrophages [24]. Different results might be due to the cell types, however, despite the same observation that oxPAPC is a new ligand directly interacting with caspase-11, the regulatory roles of oxPAPC in the activation of caspase-11 non-canonical inflammasome during the inflammatory responses are unclear and require further study.

TIR-domain-containing adapter-inducing interferon β (TRIF) is an intracellular TLR adaptor to transduce the inflammatory signal cascades from TLRs in the inflammatory cells [7]. TRIF was reported as an activator of the caspase-11 non-canonical inflammasome i.e., infection of cells with Gram-negative bacteria induced the TRIF signaling pathway in macrophages, leading to the induction and activation of caspase-11 in macrophages [25,26,27]. Although the mechanism of TRIF-induced activation of the caspase-11 non-canonical inflammasome is still unclear, TRIF in the TLR-TRIF axis is a critical molecule to activate caspase-11 non-canonical inflammasome during the inflammatory responses.

*Candida albicans* is a pathogenic yeast that induces candidiasis and secretes aspartyl proteinases, which are the key determinants for its pathogenesis. Of the several types of secreted aspartyl proteinases, secreted aspartyl proteinase 2 and secreted aspartyl proteinase 6 were reported to play a critical role in the activation of the caspase-11 non-canonical inflammasome i.e., they induced the activation of caspase-11 non-canonical inflammasome in macrophages in a type I IFN-dependent manner [28]. This study provides the insight that not only Gram-negative bacteria, but also yeast can activate the caspase-11 non-canonical inflammasome in the inflammatory responses.

Taken together, caspase-11 non-canonical inflammasome can be activated by LPS, a pathogenic component of Gram-negative bacteria, host factors, such as oxPAPC and TRIF, and other pathogenic components derived from protozoan parasite and yeast, such as LPG and secretes aspartyl proteinases. Although these molecules have been identified as the activators of caspase-11 non-canonical inflammasome during the inflammatory responses, their mechanisms of actions are different and some are still unknown. Therefore, further studies uncovering these unknown mechanisms need to be carried out. Moreover, efforts to identify novel activators of caspase-11 non-canonical inflammasome are also highly required.

### 2.3. Ligand Internalization

The primary location of the infected Gram-negative bacteria is extracellular, indicating that LPS needs to be internalized into the host cells to induce the caspase-11 non-canonical inflammasome-mediated inflammatory responses. Therefore, how LPS enters the host cells is of high interest, and several studies have reported the mechanisms of LPS internalization.

Extracellular LPS released from Gram-negative bacteria directly binds with CD14 with the help of LPS-binding protein, and, in turn, CD14 delivers LPS to MD2/TLR4, finally leading to the formation of the LPS/MD2/CD14/TLR4 complex, which is internalized by TLR4-mediated endocytosis [29]. Other types of cell surface receptors were reported to internalize extracellular LPS. Extracellular LPS directly binds with hepatocyte-related high-mobility group box 1 (HMGB1) secreted from the LPS-stimulated hepatocytes with the help of both A and B box domains of HMGB1, leading to the formation of the LPS–HMGB1 complex. The LPS–HMGB1 complex, in turn, binds with the cell surface receptor for advanced glycation end-product (RAGE) to form a LPS/HMGB/RAGE complex, and the LPS/HMGB/RAGE complex is internalized by RAGE receptor-mediated endocytosis [30]. Extracellular LPS also directly binds with secretoglobin3A2, a small protein predominantly secreted from the airway cells, leading to the formation of the LPS-secretoglobin3A2 complex. The LPS-secretoglobin3A2 complex, in turn, binds with a cell surface receptor, syndecan-1, and the domain mapping study revealed that secretoglobin3A2 of the LPS-secretoglobin3A2 complex directly binds with the heparin sulfate chains on syndecan-1 receptor [31]. The LPS/secretoglobin3A2/syndecan-1 complex is then internalized by syndecan-1 receptor-mediated endocytosis [31]. Gram-negative bacteria also form the LPS-containing outer membrane vesicles (OMVs) that are lipid vesicles released from the outer membrane of the bacteria, and the OMVs enter the host cells by endocytosis [32]. Several different mechanisms of OMV entry into the host cells were reported. Bacterial OMVs enter the host cells in a clathrin-coated pit-dependent manner with faster entry and slower cargo delivery or by caveolin-mediated endocytosis with slower entry and faster cargo delivery [33]. Bacterial OMVs enter the host cells via another way of endocytosis distinguished from the clathrin- or caveolin-mediated endocytosis. Bacterial OMVs enters the host cells via lipid raft-mediated endocytosis [33]. Interestingly, bacterial OMVs also enter the host cells in an endocytosis-independent manner but via direct membrane fusion between OMVs and host cell membranes [33]. These studies suggest that there are several different routes of LPS entry into the host cells, and the main routes of LPS entry are the cell surface receptor (TLR4, RAGE, and syndecan-1)-mediated and the non-cell surface receptor (clathrin, caveolin, and lipid raft)-mediated endocytosis. However, LPS also enters the host cells via membrane fusion that is an endocytosis-independent way.

Since extracellular LPS is internalized by endocytosis, the LPS is still in the endosomes. Therefore, host cells need to release and expose the internalized LPS in the endosomes to caspase-11 for the inflammatory responses. Guanylate-binding proteins (GBPs) is a family of GTPases. They are IFN-inducible GTPases since their expression is induced by IFNs [34]. Some studies reported that GBPs bind with endosomes containing the internalized LPS or Gram-negative bacteria and disrupt the endosomes by altering the membrane integrity, leading to the cytosolic release and exposure of LPS to caspase-11 [35,36].

The same studies suggest that extracellular LPS derived from Gram-negative bacteria is internalized in several ways, including MD2/CD14/TLR4-mediated endocytosis, OMV formation, and HMGB1/RAGE-mediated endocytosis, and the internalized LPS in endosomes is exposed with help from GBPs to cytosolic caspase-11 to induce caspase-11 non-canonical inflammasome-activated inflammatory responses. Further studies investigating the new mechanisms of extracellular LPS internalization and identifying novel factors to expose the internalized LPS to cytosolic caspase-11 are still required.

### 2.4. Caspase-11 Non-Canonical Inflammasome-Activated Inflammatory Responses

Among the molecules identified as activators of the caspase-11 non-canonical inflammasome, LPS was the first molecule identified as a direct activator of the caspase-11 non-canonical inflammasome, therefore, studies of caspase-11 non-canonical inflammasome activation have been mostly focused on LPS.

Internalized LPS is directly recognized by caspase-11, and LPS–caspase-11 complexes are oligomerized to form caspase-11 non-canonical inflammasome through the interaction of CARDs (Figure 1C) [12,14,37]. The caspase-11 non-canonical inflammasome is then activated by auto-proteolysis at the 285 aspartic acid residue, and the 254 cysteine of caspase-11 is a key amino acid for its enzymatic activity triggering auto-proteolysis [38]. Activation of caspase-11 non-canonical inflammasome induces the proteolytic activation of gasdermin D (GSDMD) at the 276 aspartic acid residue to produce the N-terminal pore-forming domain and the C-terminal autoinhibitory domain of GSDMD. The N-terminal pore-forming domain of GSDMD (N-GSDMD) moves to the cell membrane to generate N-GSDMD-mediated large barrel-shaped transmembrane pores that are 27-fold symmetric oligomers [13]. The N-GSDMD pores are 18 nm in inner diameter and allow the secretion of a number of cytoplasmic molecules to the extracellular space including pro-inflammatory cytokines, IL-1β and IL-18. N-GSDMD pores also increase in osmotic pressure, leading to an influx of extracellular water followed by cell swelling, and eventually to cell lysis known as pyroptosis, an inflammatory form of apoptosis [12,13,14,37]. Pyroptosis is a part of the anti-microbial inflammatory response. Pyroptosis eliminates the infected microbes by inducing the death of microbe-infected host cells during the inflammatory responses and also stimulates other immune cells to fight the infected microbes by secreting pro-inflammatory cytokines that contribute to inflammation in the tissues. Activation of caspase-11 non-canonical inflammasome also induces the proteolytic activation of inactive pro-caspase-1 through the activation of NLRP3 canonical inflammasome, and the active caspase-1 subsequently induces the proteolytic maturation of inactive pro-inflammatory cytokines, pro-IL-1β and pro-IL-18 to active IL-1β and IL-18, leading to the secretion of these pro-inflammatory cytokines [12,14,37].

Caspase-11 non-canonical inflammasome-mediated activation of NLRP3 canonical inflammasome is essential for the activation of caspase-1 and caspase-1-mediated downstream inflammatory responses, however, this functional crosstalk between these two inflammasomes has been poorly understood. Various recent studies reported clues about the caspase-11 non-canonical inflammasome-mediated activation of the NLRP3 canonical inflammasome. Potassium ion (K^+^) efflux is a critical determinant for the activation of NLRP3 canonical inflammasome, and caspase-11 non-canonical inflammasome activated NLRP3 canonical inflammasome by facilitating K^+^ efflux through cell membrane damage and the pores generated by N-GSDMD, bacterial pore-forming toxins, and P2X_7_ [37,39,40,41].

Although canonical inflammsomes and caspase-11 non-canonical inflammasome share the downstream inflammatory responses, such as GSDMD-mediated pyroptosis and caspase-1-mediated maturation and secretion of pro-inflammatory cytokines, unlike other types of canonical inflammasomes, caspase-11 non-canonical inflammasome is the only sensor of intracellular LPS, the most critical virulence factor of pathogenic Gram-negative bacteria in the induction of inflammatory responses. Moreover, caspase-11 non-canonical inflammasome plays an important role in bridging and inducing the LPS-mediated activation of NLRP3 inflammasome, although LPS is not an activating ligand of NLRP3 canonical inflammasome. The activation of the caspase-11 non-canonical inflammasome and caspase-11 non-canonical inflammasome-induced inflammatory responses are summarized in Figure 2.

## 3. Caspase-11 Non-Canonical Inflammasome as a Novel Target for Immunotherapy of Infectious and Inflammatory Diseases

Caspase-11 non-canonical inflammasome has been demonstrated to play pivotal roles in various infectious and inflammatory diseases, such as rheumatoid arthritis, neurodegenerative diseases, inflammatory bowel diseases, and inflammatory respiratory diseases [14], and selective targeting of the caspase-11 non-canonical inflammasome has been considered as a promising strategy for treating these diseases.

As discussed earlier, caspase-11 non-canonical inflammasome is activated by auto-proteolysis at the 285 aspartic acid residues, which leads to the idea that blockade of this auto-proteolytic site could suppress the activation of the caspase-11 non-canonical inflammasome. Indeed, caspase-11-processing dead knock-in (KI) mice (*Casp11^D285A/D285A^*) abolished caspase-11 auto-proteolysis and the subsequent caspase-11 non-canonical inflammasome-induced inflammatory responses in the bone- marrow-derived macrophages of these mice [38].

The 254 cysteine of caspase-11 is a key amino acid for its enzymatic activity in inducing auto-proteolysis [38]. Therefore, inhibiting the enzymatic activity of caspase-11 could be another strategy to suppress the activation of the caspase-11 non-canonical inflammasome. Caspase-11 enzymatically- dead KI mice (*Casp11^C254A/C254A^*) suppressed the activation of the caspase-11 non-canonical inflammasome and attenuated the subsequent caspase-11 non-canonical inflammasome-induced inflammatory responses in the bone-marrow-derived macrophages of these mice [38].

The caspase-11 non-canonical inflammasome is activated by direct interaction with intracellular LPS, therefore, the inhibitors interfering with the interaction between caspase-11 and LPS could inhibit the caspase-11 non-canonical inflammasome-induced inflammatory responses. Since caspase-11 interacts with LPS through its CARD, dominant-negative that has only CARD could be an effective inhibitor of the caspase-11 non-canonical inflammasome. As discussed earlier, LPS should be internalized to interact with caspase-11 and in turn, activate caspase-11 non-canonical inflammasome, therefore, blockade of LPS internalization could be a good strategy to inhibit caspase-11 non-canonical inflammasome. LPS is internalized by TLR- or RAGE-mediated endocytosis by forming LPS/MD2/CD14/TLR4 and HMGB1/RAGE/LPS complexes, which provide the idea that inhibiting LPS endocytosis by blocking the interaction of LPS with TLR and RAGE. Also, the formation of receptor complexes could attenuate LPS internalization and the activation of caspase-11 non-canonical inflammasome. Indeed, the antagonistic HMGB1 antibody blocked the interaction between HMGB1/LPS complex and RAGE and the subsequent endocytosis of the HMGB1/RAGE/LPS complex, resulting in the inhibition of caspase-11 non-canonical inflammasome [42].

Taken together, caspase-11 non-canonical inflammasome could be targeted by inhibiting auto-proteolysis, enzymatic activity, interaction with its activating ligand, LPS, and LPS internalization, as summarized in Figure 3. In addition, selective and effective targeting of caspase-11 non-canonical inflammasome by downregulating its expression via its specific small interfering RNA or CRISPR/Cas9 technology could be a promising strategy to treat infectious and inflammatory diseases, including rheumatoid arthritis, infectious diseases caused by Gram-negative bacteria, such as *Citrobacter rodentium*, *Salmonella typhimurium*, *Burkholderia thailandensis*, and Legionella pneumophila, neurodegenerative diseases, such as multiple sclerosis, amyotrophic lateral sclerosis, and Parkinson’s disease, inflammatory bowel diseases, such as colitis, and inflammatory respiratory disease, such as chronic obstructive pulmonary disease since caspase-11 non-canonical inflammasome plays a critical role in inducing inflammatory responses in these infectious and inflammatory diseases [14].

## 4. Conclusion and Perspectives

Inflammation is an immune response to protect the body from pathogen infection and cellular danger signals. However, chronic inflammation has been thought of as a major risk factor for many human diseases, such as infectious, inflammatory, and autoimmune diseases, and even cancer. Therefore, efforts have been made to understand the mechanisms of inflammatory responses and develop effective anti-inflammatory therapeutics to treat these diseases.

Triggering, as one of the steps of inflammatory responses, is an inflammation-boosting step that shows inflammasome activation as a cardinal feature, therefore, many studies have demonstrated that inflammasomes play crucial roles in inflammatory responses and the pathogenesis of various diseases. Recently, new inflammasomes activated in response to a novel ligand, an intracellular LPS were discovered. These are known as non-canonical inflammasomes since they are different from canonical inflammasomes in the activating ligand and the underlying activation mechanism. Once host cells are infected with Gram-negative bacteria, extracellular LPS derived from the Gram-negative bacteria enters the host cells by receptor-mediated endocytosis and, in turn, directly interacts with caspase-11, leading to the formation and activation of the caspase-11 non-canonical inflammasome, which subsequently induces two main inflammatory responses: 1) GSDMD proteolytic processing and pyroptosis by generating N-GSDMD-mediated membrane pores and 2) caspase-1 proteolytic activation and caspase-1-mediated proteolytic maturation and secretion of pro-inflammatory cytokines, IL-1β and IL-18.

LPS is one of the strongest pathogenic factors of Gram-negative bacteria to induce inflammatory responses, and caspase-11 non-canonical inflammasome is the only molecular platform to induce inflammatory responses in response to Gram-negative bacterial LPS. Therefore, targeting caspase-11 non-canonical inflammasome is a new and feasible therapeutic strategy for infectious and inflammatory diseases, and it is distinguished from targeting canonical inflammsomes or other inflammatory molecules that are already known. Indeed, some strategies, such as the inhibition of caspase-11 enzymatic activity and the blockade of the caspase-11 auto-proteolysis site successfully attenuated the activation of caspase-11 non-canonical inflammasome during Gram-negative bacterial infection. Another strategy is inhibiting LPS endocytosis by blocking the interaction of LPS with its receptor, RAGE, which suppresses the activation of the caspase-11 non-canonical inflammasome. Therefore, selective and effective targeting of caspase-11 non-canonical inflammasome could be a new and promising approach to treat infectious and inflammatory diseases. However, despite these successful studies, other strategies inhibiting caspase-11 non-canonical inflammasome discussed in Section 3 need to be further investigated and practically demonstrated. Additionally, more studies, including (1) identifying new molecules that play critical roles in the activation of caspase-11 non-canonical inflammasome, (2) understanding new molecular mechanisms by which caspase-11 non-canonical inflammasome activates inflammatory responses, and (3) demonstrating new strategies targeting caspase-11 non-canonical inflammasome are highly demanded.

In conclusion, given the evidence, caspase-11 non-canonical inflammasome is a key determinant of inflammatory responses, suggesting that selective and effective targeting of caspase-11 non-canonical inflammasome could be a promising strategy for developing novel anti-inflammatory therapeutics that can treat infectious and inflammatory diseases.

## Figures and Tables

**Figure 1 ijms-21-02736-f001:**
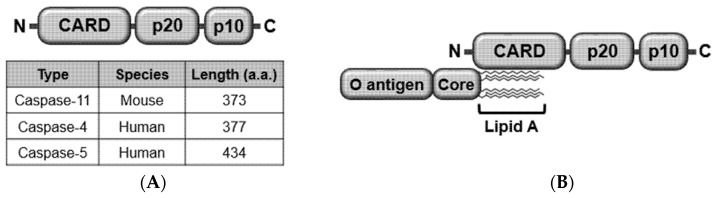
Structure and activation of caspase-11 non-canonical inflammasome. (**A**) Structures of mouse caspase-11 and human caspase-4 and -5. The three non-canonical inflammasomes have the same domain structures consisting of an N-terminal CARD, a p20, and C-terminal p10, but different sizes as indicated. (**B**) LPS directly interacts with mouse caspase-11 or human caspase-4/5 through a lipid A moiety of LPS and a CARD of caspase-11. (**C**) Activation of caspase-11 non-canonical inflammasome by oligomerizing LPS-caspase-4/5/11 complexes. CARD, caspase recruit domain; LPS, lipopolysaccharide.

**Figure 2 ijms-21-02736-f002:**
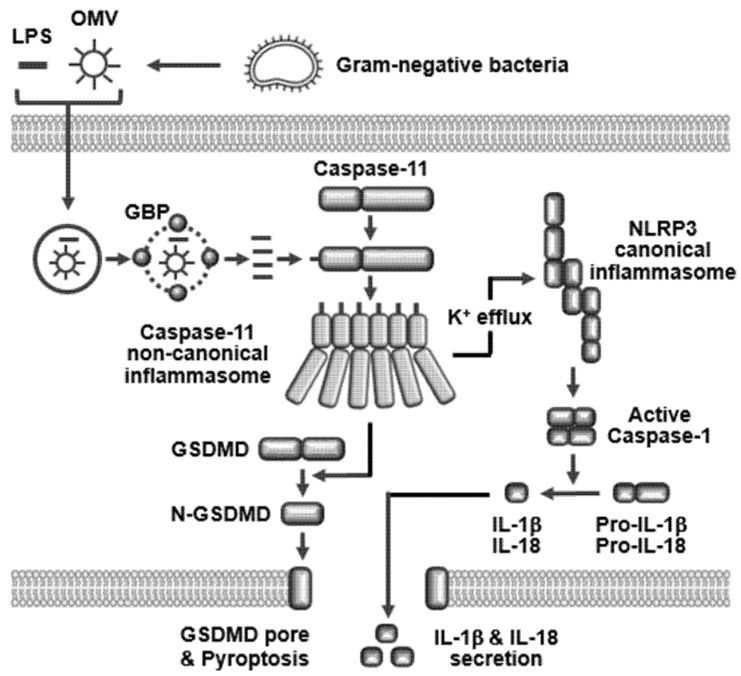
Caspase-11 non-canonical inflammasome-induced inflammatory responses. LPS or OMVs derived from Gram-negative bacteria enter host cells via several different routes (receptor-mediated endocytosis, non-receptor-mediated endocytosis, and membrane fusion) and are released from the endosomes with help of GBPs, leading to the direct interaction with caspase-11 to form LPS–caspase-11 complexes. Caspase-11 non-canonical inflammasome is activated by oligomerization of the LPS–caspase-11 complexes. Activation of caspase-11 non-canonical inflammasome induces proteolysis of the full length of GSDMD generating the N-GSDMD pore-forming domain, which migrates to cell membranes to produce the N-GSDMD-mediated membrane pores. N-GSDMD pores induce osmotic pressure via water influx, cell swelling, and pyroptosis. On the other hand, activated caspase-11 non-canonical inflammasome activates NLRP3 canonical inflammasome via facilitating potassium ion efflux, which subsequently induces proteolytic activation of caspase-1. Active caspase-1 subsequently induces the proteolytic maturation of inactive pro-inflammatory cytokines, pro-IL-1β, and pro-IL-18 to active IL-1β and IL-18, leading to the secretion of these active IL-1β and IL-18 through the N-GSDMD pores. OMV, outer membrane vesicle; GSDMD, gasdermin D; N-GSDMD, N-terminal domain of GSDMD.

**Figure 3 ijms-21-02736-f003:**
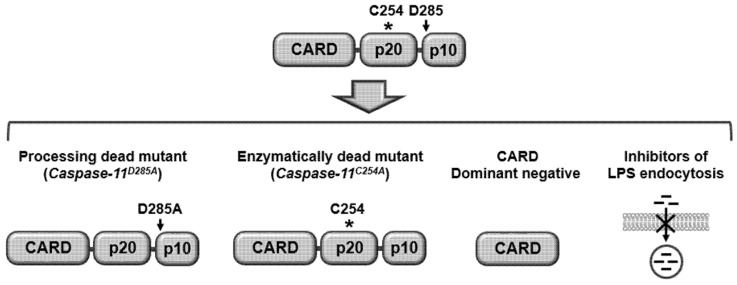
Potential strategies targeting caspase-11 non-canonical inflammasome. Activation of caspase-11 non-canonical inflammasome can be inhibited by generating caspase-11 processing-dead mutant (Caspase-11D285A) and caspase-11 enzymatically-dead mutant (Caspase-11C254A). Also, by preventing the interaction between LPS and caspase-11 using CARD dominant-negative and the LPS internalization via blocking the interaction of LPS with its receptors, such as TLR4 and RAGE. TLR4, Toll-like receptor 4; RAGE, receptor for advanced glycation end-product. *: Caspase-11 enzymatic activity site, arrow: Caspase-11 auto-proteolysis site

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
