# Peer review of "Caspase-11 Non-Canonical Inflammasome: Emerging Activator and Regulator of Infection-Mediated Inflammatory Responses"

_ijms, 2020, doi:10.3390/ijms21082736_

Round 1

Reviewer 1 Report

The authors present a comprehensive review of Caspase-11 non-canonical inflammasome.

Specific comments are below:

All Greek letters are missing.

Line 47. Many experts would consider pyroptosis a different mechanism than apoptosis. Please revise the statement.

Line 80. Typo, should it be “activators?”

Abbreviations should be kept at a minimum. Some sentences are like a soup of letters. Please spell these words, as sometimes they are only repeated in one single paragraph, and there is no need to use an abbreviation: NLR, RIG RLrS, AIM2, ALRS, GSDMD, LPG, OxPac, Saps, SCBG3A1, GAGE.

Line 230. Please explain in which model.

Line 255 Please give examples.

The authors should include a section of the modulation of the inflammasome. What happens after it is activated? What is the negative feedback if any?

Author Response

1. All Greek letters are missing.

- Thank you for your comments, it was due to the font of the manuscript. All Greek letters were appropriately replaced in all parts of the manuscript.

2. Line 47. Many experts would consider pyroptosis a different mechanism than apoptosis. Please revise the statement.

- This is a good point. As you pointed out, pyroptosis is different with apoptosis. Therefore, the statement “pyroptosis (an inflammatory form of apoptosis)” was revised to “pyroptosis (an inflammatory form of cell death)”. An inflammatory form of cell death is the expression that generally describes pyroptosis and is used in many papers.

3. Line 80. Typo, should it be “activators?”

- It is my mistake. It should be ‘Activators’ and was corrected to ‘Activators’.

4. Abbreviations should be kept at a minimum. Some sentences are like a soup of letters. Please spell these words, as sometimes they are only repeated in one single paragraph, and there is no need to use an abbreviation: NLR, RIG RLrS, AIM2, ALRS, GSDMD, LPG, OxPac, Saps, SCBG3A1, GAGE.

- Thank you for your comment. Unnecessary abbreviations were spelled in full names, including the abbreviations you pointed out. However, several abbreviations, such as NLR, GSDMD, LPG, oxPAPC, RAGE were used many times in many parts of the manuscript, therefore, we still used these abbreviations in the manuscript.

5. Line 230. Please explain in which model.

- These results were in the bone marrow-derived macrophages in the caspase-11-processing dead knock-in (KI) mice, and the model was specified in the manuscript. Additionally, the model was also specified for another result of the same study (line 275).

6. Line 255 Please give examples.

- ‘Targeting of caspase-11 non-canonical inflammasome’ in the manuscript means downregulating its expression. Therefore, the statement was revised including the examples as below;

“In addition, selective and effective targeting of caspase-11 non-canonical inflammasome by downregulating its expression vis its specific small interfering RNA or CRISPR/Cas9 technology could be a promising strategy to treat infectious and inflammatory diseases.”

7. The authors should include a section of the modulation of the inflammasome. What happens after it is activated? What is the negative feedback if any?

- This is a good point. The biological events that happens after the activation of caspase-11 non-canonical inflammasome were described in the ‘2.4. Caspase-11 non-canonical inflammasome-activated inflammatory responses’ section and also summarized in the Fig. 2. Regarding your comment of the negative feedback after the activation of caspase-11 non-canonical inflammasome, unfortunately, no specific study has been reported, so far. However, it is well studied and established that the activation of caspase-11 non-canonical inflammasome eventually results in the pyroptosis, an inflammatory form of cell death, leading to the breaking down and degradation of caspase-11 non-canonical inflammasome in the pyroptotic cells. This could be considered as the negative feedback after the activation of caspase-11 non-canonical inflammasome.

Reviewer 2 Report

In the Review: “Caspase-11 non-canonical inflammasome: Emerging activator and regulator of infection-mediated inflammatory responses”, the Author intended:

  • To describe the recent progress in non-canonical inflammasome roles in inflammation, focusing on caspase-11 non-canonical inflammasome.
  • To highlight novel therapeutic approaches for the prevention and treatment of infectious and inflammatory diseases by targeting caspase-11 non-canonical inflammasome.

Although the aim of the Review concerns a very interesting and ongoing topic, the Review is not clearly written and sometimes difficult to understand. The Author does not deeply and clearly describe non-canonical inflammasome mechanism of activation and its relative role in inflammation. The Author does not explain properly why caspase-11 non canonical inflammasome plays a pivotal role in a wide range of inflammatory diseases and, therefore, it is difficult for the reader to understand the importance of new therapeutic approaches targeting this molecular platform.

  1. Introduction:

Introduction is not clearly written and sometimes difficult to understand (for example it looks like that NLR inflammasomes are non-canonical ones) and mainly focused on inflammation and canonical inflammasomes instead of non-canonical ones.

  1. Role of caspase-11 non canonical inflammasome

The title of  this section does not appear correct because the role of caspase-11 non canonical inflammasome has not been described.

Activators section is not clearly written and it is very difficult to understand the different mechanism of action of the reported activators. Ligand Internalization section is not clearly written and not focused on the importance of the different internalization mechanisms on non-canonical activation.

In Caspase-11 non canonical inflammasome-activated inflammatory responses section, it is not clearly addressed the importance of non-canonical inflammasome in inflammation. In addition, it is not clearly addressed gasdermin D mechanism of action in pyroptosis and the importance of this kind of cell death in inflammation. Moreover,  it is not clearly explained potassium ion efflux critical role in the cross-talk between caspase-11 and NLRP3 inflammasome. Legend of Figure 2 is not clearly written.

  1. Caspase-11 non canonical inflammasome as a novel target for immunotherapy of infectious and inflammatory diseases

In this section the Author does not explain why caspase-11 non canonical inflammasome plays a pivotal role in infectious and inflammatory diseases and subsequently for the reader it is difficult to understand the importance of new therapeutic approaches targeting this molecular platform.

Author Response

To describe the recent progress in non-canonical inflammasome roles in inflammation, focusing on caspase-11 non-canonical inflammasome.

To highlight novel therapeutic approaches for the prevention and treatment of infectious and inflammatory diseases by targeting caspase-11 non-canonical inflammasome.

Although the aim of the Review concerns a very interesting and ongoing topic, the Review is not clearly written and sometimes difficult to understand. The Author does not deeply and clearly describe non-canonical inflammasome mechanism of activation and its relative role in inflammation. The Author does not explain properly why caspase-11 non canonical inflammasome plays a pivotal role in a wide range of inflammatory diseases and, therefore, it is difficult for the reader to understand the importance of new therapeutic approaches targeting this molecular platform.

1. Introduction:

Introduction is not clearly written and sometimes difficult to understand (for example it looks like that NLR inflammasomes are non-canonical ones) and mainly focused on inflammation and canonical inflammasomes instead of non-canonical ones.

 - Introduction section was structured of 1) the concept of inflammation, 2) two types of inflammatory responses – priming and triggering, and 3) the introduction of canonical and non-canonical inflammasomes. NLR inflammasomes, absent in melanoma 2 (AIM2) inflammasome, and pyrin inflammasome are the canonical inflammasomes. To avoid confusion, this paragraph (paragraph 4) was re-written to make the meaning of description clear. Additionally, the description of non-canonical inflammasomes was more added in the paragraph 4 and 5 of the Introduction section, and the specific description of non-canonical inflammasomes was described in the following section of the manuscript.

2. Role of caspase-11 non canonical inflammasome

The title of this section does not appear correct because the role of caspase-11 non canonical inflammasome has not been described.

 - Thank you for your comments. I tried to describe the structures, activation and the activation-mediated molecular mechanisms (roles) of caspase-11 non-canonical inflammasome in this section. It is generally considered that the roles of caspase-11 non-canonical inflammasome activation include the GSDMD processing and GSDMD-induced pyroptosis via generating GSDMD pores in cell membranes as well as the caspase-1 processing and caspase-1-induced maturation and secretion of pro-inflammatory cytokines. Therefore, I titled this section ‘Role of caspase-11 non-canonical inflammasome’. As your recommendation, the tile was changed to ‘Activation of caspase-11 non-canonical inflammasome’.

3. Activatorssection is not clearly written and it is very difficult to understand the different mechanism of action of the reported activators.

- Thank you for your comment. LPS was first discovered as an activator of the caspase-11 non-canonical inflammasome and most of the previous studies focused on LPS. Therefore, the mechanism by which LPS activates the caspase-11 non-canonical inflammasome is well established and easily understandable, so that the mechanism of LPS-induced activation of the caspase-11 non-canonical inflammasome was described in the manuscript in detail. However, the other activators, such as LPG, oxPAPC, TRIF, and Saps were recently discovered as potential activator of the caspase-11 non-canonical inflammasome, and the mechanisms by which these molecules activate the caspase-11 non-canonical inflammasome have been poorly understood, so far. Therefore, only introduction of these molecules as potential activators of the caspase-11 non-canonical inflammasome was described, and the detailed mechanisms of the caspase-11 non-canonical inflammasome activated by these molecules were not clearly described in this section of the manuscript.

4. Ligand Internalizationsection is not clearly written and not focused on the importance of the different internalization mechanisms on non-canonical activation.

- Thank you for your comment. Each LPS internalization mechanism was re-described in more detail. Additionally, difference in LPS internalization mechanisms was also described and compared in this section.

5. In Caspase-11 non canonical inflammasome-activated inflammatory responses section, it is not clearly addressed the importance of non-canonical inflammasome in inflammation. In addition, it is not clearly addressed gasdermin D mechanism of action in pyroptosis and the importance of this kind of cell death in inflammation. Moreover, it is not clearly explained potassium ion efflux critical role in the cross-talk between caspase-11 and NLRP3 inflammasome. Legend of Figure 2 is not clearly written.

 - The importance of non-canonical inflammasome in inflammatory responses is described at the end of this section.

- Gasdermin D mechanism of action in pyroptosis and the importance of pyroptosis in inflammatory responses are described in more detail in this section.

- It is well-known that potassium ion efflux activates NLRP3 inflammasome, however, the underlying molecular and cellular mechanisms of potassium ion efflux-mediated NLRP3 inflammasome activation are still unknown. Interestingly, although the mechanism is also unknown, caspase-11 has been reported to induce potassium ion efflux, therefore, it was described in the manuscript that potassium ion efflux critical role in the cross-talk between caspase-11 and NLRP3 inflammasome. Since the mechanisms of caspase-11-induced potassium ion efflux and potassium ion efflux-induced NLRP3 inflammasome activation are still unknown, the studies in these regards are highly required.

- As your recommendation, Fig. 2 legend was re-written more clearly.

6. Caspase-11 non canonical inflammasome as a novel target for immunotherapy of infectious and inflammatory diseases

In this section the Author does not explain why caspase-11 non canonical inflammasome plays a pivotal role in infectious and inflammatory diseases and subsequently for the reader it is difficult to understand the importance of new therapeutic approaches targeting this molecular platform.

- The pivotal roles of caspase-11 non-canonical inflammsome in the infectious and inflammatory diseases, such as infectious diseases induced by various pathogens, neurodegenerative diseases, inflammatory bowel diseases, rheumatoid arthritis, and inflammatory respiratory diseases were described in detail by discussing a number of studies investigating the activation of caspase-11 non-canonical inflammasome in these infectious and inflammatory diseases in the previous study (Yi, 2018), therefore, I stated in the manuscript that “caspase-11 non-canonnical inflammasome has been demonstrated to play pivotal roles in various infectious and inflammatory diseases” by citing this previous study (Yi , 2018).

- LPS is the most critical virulence factor of pathogenic Gram-negative bacteria to induce the inflammatory responses, and caspase-11 non-canonical inflammasome is the only molecular platform to induce the inflammatory responses in response to Gram-negative bacterium-derived LPS. Therefore, targeting caspase-11 non-canonical inflammasome is new and feasible therapeutic strategy that is distinguished form targeting canonical inflmmsomes or other inflammatory molecules that are already known. Therefore, selective and effective targeting caspase-11 non-canonical inflammasome could be a new and promising approach to treat infectious and inflammatory diseases. This statement was included in the 4. Conclusion and perspectives section and the 4. Conclusion and perspectives section was revised.

Round 2

Reviewer 1 Report

Response to items to the reviewer should make into the manuscript itself.

See examples below:

Previous point 6: The review asked for "examples." The author's response is below but it remains unanswered. Please give specific examples of infectious and inflammatory diseases. Otherwise, the statement is too generic. Can the modulation of caspase 11 make a difference in pneumonia? cellulitis? cold?

"n addition, selective and effective targeting of caspase-11 non-canonical inflammasome by downregulating its expression vis its specific small interfering RNA or CRISPR/Cas9 technology could be a promising strategy to treat infectious and inflammatory diseases."

Previous point 7:

7. The authors should include a section of the modulation of the inflammasome. What happens after it is activated? What is the negative feedback if any?

Author's response: This is a good point. The biological events that happens after the activation of caspase-11 non-canonical inflammasome were described in the ‘2.4. Caspase-11 non-canonical inflammasome-activated inflammatory responses’ section and also summarized in the Fig. 2. Regarding your comment of the negative feedback after the activation of caspase-11 non-canonical inflammasome, unfortunately, no specific study has been reported, so far. However, it is well studied and established that the activation of caspase-11 non-canonical inflammasome eventually results in the pyroptosis, an inflammatory form of cell death, leading to the breaking down and degradation of caspase-11 non-canonical inflammasome in the pyroptotic cells. This could be considered as the negative feedback after the activation of caspase-11 non-canonical inflammasome.

Reveiwer's comment: please add a paragraph on the review about the negative regulation of inflammasome. There are reports showing that IFN-gamma can shut down the inflammasome. While there is much interest on knowing how things are activated, deactivation or disassembly of the inflammasome must be also a tightly regulated event. I suggest the authors searching for it in the literature and when formulating your reply, please add i to the review itself and not only to reviwer's eyes.

Author Response

Previous point 6: The review asked for "examples." The author's response is below but it remains unanswered. Please give specific examples of infectious and inflammatory diseases. Otherwise, the statement is too generic. Can the modulation of caspase 11 make a difference in pneumonia? cellulitis? cold?

"In addition, selective and effective targeting of caspase-11 non-canonical inflammasome by downregulating its expression vis its specific small interfering RNA or CRISPR/Cas9 technology could be a promising strategy to treat infectious and inflammatory diseases."

 - I misunderstood your point and thought that you asked to provide the examples of the ways of targeting, not the diseases. I re-stated this sentence including the specific infectious and inflammatory diseases with a corresponding reference (line 292 - 300), as below;

“In addition, selective and effective targeting of caspase-11 non-canonical inflammasome by downregulating its expression via its specific small interfering RNA or CRISPR/Cas9 technology could be a promising strategy to treat infectious and inflammatory diseases, including rheumatoid arthritis, infectious diseases caused by Gram negative bacteria, such as C. rodentium, S. typhimurium, B. thailandensis, and L. pneumophilia, neurodegenerative diseases, such as multiple sclerosis, amyotrophic lateral sclerosis, and Parkinson’s disease, inflammatory bowel diseases, such as colitis, and inflammatory respiratory disease, such as chronic obstructive pulmonary disease since caspase-11 non-canonical inflammasome plays a critical role to induce inflammatory responses in these infectious and inflammatory diseases [14]”

Previous point 7:

7. The authors should include a section of the modulation of the inflammasome. What happens after it is activated? What is the negative feedback if any?

Author's response: This is a good point. The biological events that happens after the activation of caspase-11 non-canonical inflammasome were described in the ‘2.4. Caspase-11 non-canonical inflammasome-activated inflammatory responses’ section and also summarized in the Fig. 2. Regarding your comment of the negative feedback after the activation of caspase-11 non-canonical inflammasome, unfortunately, no specific study has been reported, so far. However, it is well studied and established that the activation of caspase-11 non-canonical inflammasome eventually results in the pyroptosis, an inflammatory form of cell death, leading to the breaking down and degradation of caspase-11 non-canonical inflammasome in the pyroptotic cells. This could be considered as the negative feedback after the activation of caspase-11 non-canonical inflammasome.

Reviewer's comment: please add a paragraph on the review about the negative regulation of inflammasome. There are reports showing that IFN-gamma can shut down the inflammasome. While there is much interest on knowing how things are activated, deactivation or disassembly of the inflammasome must be also a tightly regulated event. I suggest the authors searching for it in the literature and when formulating your reply, please add it to the review itself and not only to reviewer's eyes.

- Thank you for your comment. This article is focused mainly on caspase-11 non-canonical inflammasome, not broad range of inflammasomes, such as various canonical inflammasomes. I found several papers reporting the modulation of canonical inflammasomes by IFN, but based on previous studies, IFN-gamma does not suppress the activation of caspase-11 non-canonical inflammasome during Gram-negative bacterial infection. Contrary, caspase-11 non-canonical inflammasome activation requires IFN priming for effective activation against Gram-negative bacteria, and studies showed that caspase-11 expression was IFN-β or IFN-γ inducible in mouse macrophages although a basal level of caspase-11 expression appeared to be sufficient for activation by intracellular LPS (please refer below 1 – 3 literatures). Moreover, Guanylate-binding protein (GBP) is essential for LPS release from the endosomes by degrading endosomes, and the expression of GBP is induced by IFN-gamma.

As you pointed out, negative regulation of caspase-11 non-canonical inflammasome is much interest, however, it is still unclear how caspase-11 non-canonical inflammasome including human caspase-4/5 non-canonical inflammasomes, not other types of canonical inflammasomes, such as NLRP1, NLRP3, NLRC4, NLRP6, AIM2, pyrin inflammasomes are negatively regulated after activation, and further studies in this regard are highly required.

1) Schauvliege et al., Caspase-11 gene expression in response to lipopolysaccharide and interferon- requires nuclear factor-κB and signal transducer and activator of transcription (STAT) 1. J Biol Chem. 2002; 277: 41624–41630

2) Rathinam et al., TRIF licenses caspase-11-dependent NLRP3 inflammasome activation by gram-negative bacteria. Cell. 2012; 150: 606–619

3) Kayagaki et al., Noncanonical Inflammasome Activation by Intracellular LPS Independent of TLR4. Science. 2013; 341: 1246–1249

Reviewer 2 Report

The revised version of the Review “Caspase-11 non-canonical inflammasome: Emerging activator and regulator of infection-mediated inflammatory responses” provided by the Author has been significantly improved in all its sections.

Author Response

I appreciate your insightful and helpful comments to improve the quality of this article.